# Occurrence of shigellosis in pediatric diarrheal patients in Chattogram, Bangladesh: A molecular based approach

A. K. M. Zakir Hossain[1]⊛, Md. Zahid Hasan[1], Sohana Akter Mina[1], Nahid Sultana[2], A. M. Masudul Azad Chowdhury[1]⊛ *

1 Department of Genetic Engineering & Biotechnology, Laboratory of Microbial & Cancer Genomics, University of Chittagong, Bangladesh, 2 Department of Microbiology, Chattogram Maa-O-Shishu Hospital Medical College, Chattogram, Bangladesh

⊛ These authors contributed equally to this work.
* masud.geb@cu.ac.bd

**Data Availability Statement:** All relevant data are within the paper.

**Funding:** The authors received financial support from University Grant Commission, Bangladesh for

## Abstract

*Shigella* a Gram-negative, non-motile bacillus, is the primary causative agent of the infectious disease shigellosis, which kills 1.1 million people worldwide every year. The children under the age of five are primarily the victims of this disease. This study has been conducted to assess the prevalence of shigellosis through selective plating, biochemical test and conventional PCR assays, where the samples were collected from suspected diarrheoal patients. Invasive plasmid antigen H (*ipaH*) and O-antigenic *rfc* gene were used to identify *Shigella spp.* and *S. flexneri* respectively. For validation of these identification, PCR product of *ipaH* gene of a sample (*Shigella flexneri* MZS 191) has been sequenced and submitted to NCBI database (GenBank accession no- MW774908.1). Further this strain has been used as positive control. Out of 204, around 14.2% (n = 29)(P> 0.01) pediatric diarrheoal cases were screened as shigellosis. Another interesting finding was that most of shigellosis affected children were 7 months to 1 year (P> 0.01).The significance of this study lies in the analyses of the occurrence and the molecular identification of *Shigella spp.* and *S. flexneri* that can be utilized in improving the accurate identification and the treatment of the most severe and alarming shigellosis.

## Introduction

The diarrhea is the major symptom of shigellosis which is an acute intestinal infection [1] caused by *Shigella* [2]. These bacteria belong to the group of heat-resistant, Gram-negative, facultative, human host-specific pathogen and have the ability to invade and infect the cells that make up the intestinal lining [3]. Shigellosis ranges from mild watery diarrhea to severe bacillary dysentery with fever, abdominal pain, and frequent passage of bloody, mucoid and small-volume stools [4,5]. High fever, strong abdominal cramps, painful frequent passage of small-volume stools consisting of blood, mucus, inflammatory cells, and fecal matter are the symptoms of shigellosis [6,7]. Shigellosis leads to death and constitutes one of the major morbidities

conducting this research. Fund Reference Number: 37-01-0000-073-04-013/2019/1750. funders had no role in study design, data collection and analysis, decision to publish, or preparation of the manuscript, http://www.ugc.gov.bd

**Competing interests:** The authors have declared that no competing interests exist

and mortalities among infant and children worldwide [8]. It was estimated that worldwide *Shigella* causes 164.7 million cases of shigellosis with 163.2 million from developing countries resulting 1.1 million deaths annually [9]. Recently in Asia, it was reported that the number of shigellosis cases were at nearly 91 million, resulting in 414,000 deaths each year [10].There is no exact clear demography, however, some previous studies conducted in six- Asian countries (Bangladesh, Pakistan, Thailand, China, Vietnam and Indonesia), the burden of *Shigella* infection rates were high in children under the age of 5 [11].

There are four species of *Shigella* based on antigenic properties: *S. dysenteriae, S. flexneri, S. boydii and S. sonnei* [12]. Among these,*Shigella flexneri* is mostly isolated in developing countries [7]. In Bangladesh, *S. flexneri* is the most frequently isolated (54%-60%) gastrointestinal pathogen [13] and it has been estimated that annually more than 95,000 children<5years of age die due to shigellosis [5,14]. *Shigella flexneri* is highly infectious and $10^2-10^3$ bacteria is suffice to cause diarrhea in human [15,16]. This low infective dose has ability to survive in low acidic environment in human stomach by up regulating the acid resistance genes [17]. *Shigellaflexneri* invades the colonic and rectal epithelium of primates and humans not through apical surface, it uses the antigen sampling microfold cells (M cells) and basolateral surface of its target cells, causing the acute mucosal inflammation characteristic of shigellosis [18]. Now shigellosis is a major public health concern in low socioeconomic countries [1] like Bangladesh where people live in poor sanitation and in overcrowded condition [16].

The epidemiological data of Shigellosis in Bangladesh perspective and the proper screening of *Shigella spp* as well as *Shigella flexneri* are needed to conduct a proper treatment. Timely and accurate diagnosis of enteric diseases and identification of their causative agents are essential for effective patient management and appropriate treatment. Such measures are essential for infection control and addressing the significant public health challenges associated with this disease [19]. The traditional methods based on culture techniques, biochemical, and serological identification for detecting Shigellosis are associated with long processing times, demanding workload and have relatively low specificity and sensitivity [3,20]. In addition, detecting *Shigella spp.* is restricted to highly contagious environments and suboptimal preservation of the sample may lead to erroneous outcomes, including false positives or negatives.To mitigate these issues, numerous molecular-based approaches like PCR assay have been proposed [3]. The PCR-based identification of microorganisms like *Shigella spp.* are considered to be rapid, highly sensitive, specific, and reliable, making them a preferable alternative to the traditional method [2,20]. The effector invasive plasmid antigen H (*ipaH*) gene [20] and other invasive plasmid antigens (*ipa*) can be used as the potential target for molecular identification of *Shigella* [21]. The IpaH is a unique polypeptide that is found only in *Shigella* and *EnteroinvasiveEscherichia coli* (*EIEC*) [21]. On the other hand, the O-antigenic *rfc* gene is excellent candidate for *S. flexneri* identification among *Shigella spps.* as this gene is specific for *S. flexneri* [7]. It was included to confirm a rapid, accurate and a reliable method for identification of *Shigella* and *S. flexneri*. After accurate identification, the prevalence of Shigellosis caused by *Shigella spp.* as well as *Shigella flexneri* in the case of pediatric diarrheoal patients in Chattogram, Bangladesh was evaluated.

## Material and methods

### Sample collection

In the current study, human clinical samples were investigated therefore, ethical approval was obtained from the "Institutional Review Board" of Chattogram Maa-O-Shishu Hospital Medical College, Chattogram, Bangladesh (Ref: CMOSHMC/IRB/2020/03). The guardians of pediatric patients gave written consent to the hospital authority, and the Institutional Review

Board (IRB) waived additional consent form and approved the use of these samples for our research. Stool samples were collected from the diarrheal word, Chittagong Maa-O-Shishu Hospital Medical College and stored at -20°C in our 'Laboratory of Microbial and Cancer Genomics', Department of Genetic Engineering and Biotechnology, University of Chittagong, Chattogram, Bangladesh.

## Sample culture

The samples were cultured on XLD agar (Xylose Lysine Deoxycholate) medium for primary screening. Then the pink colonies were transferred on MacConkey agar for pure culture and transparent colonies were suspected for *Shigella flexneri*.

## Biochemical test

The triple sugar iron (TSI) agar medium was used for biochemical test. The inoculated TSI test tubes were incubated at 37°C for overnight. The TSI test tubes within alkaline slant and acidic butt without gas and $H_2S$ were suspected for *Shigella spp*. and *S. flexneri*. These suspected samples were selected for DNA isolation and molecular identification. Other TSI test tubes (alkaline slant + acidic butt with gas or $H_2S$, alkaline slant + alkaline butt, acidic slant + acidic butt with and without gas or crack) were rejected.

## Suspected sample storage

The nutrient agar slant was used for short-term storage and 10% skim milk was used for long-term storage.

## DNA extraction

This study used the boiling method to isolate bacterial DNA [22,23]. After purity and DNA concentration check through NanoDrop, highly purified and concentrated DNA samples were selected for PCR amplification and stored at -20°C.

## Molecular identification of *Shigella spp*. and *S.flexneri*

The primer pairs of *ipaH* and primer pairs of *rfc* gene (Table 1) were designed. PCRs were performed in a 10 μl mixture, composed of 5 μl of PCR master mix (Promega), 0.5 μl forward, 0.5 μl reverse primers (10 mmol/ml), 3 μl nuclease free water and 1 μl of sample DNA template. For *ipaH* gene identification, PCR amplification reactions were performed in a PCR machine (Nyx Technik, Inc. model: A6, USA) under the following conditions: 95°C for 15 min and then 33 cycles of 15 S at 95°C, 30 S at 58°C, 30 S at 70°C and one cycle final elongation was performed for 7 min at 72°C. For *rfc* gene identification, PCR amplification conditions were 95°C for 3 min and then 35 cycles of 35 S at 95°C, 35 S at 48°C, 25 S at 72°C and one cycle final elongation was performed for 2 min at 72°C. Amplified PCR products were

**Table 1. List of primers.**

| Gene | Primer (5'>3') | Amplicon size | GenBank accession no. | Reference article |
|------|----------------|---------------|----------------------|-------------------|
| *ipaH* | F: 5'-ACCATGCTCGCAGAGAAACT-3' | 181 | M7644 | [21] |
|  | R: 5'-TACGCTTCAGTACAGCATGC-3' |  |  |  |
| *rfc* | F 5'-TTT ATG GCT TCT TTG TCG GC-3' | 537 | CP000266 | [24] |
|  | R 5'-CTG CGT GAT CCG ACC ATG-3' |  |  |  |

visualized and verified by separating and comparing the DNA bands with appropriate DNA ladder and positive control (*Shigella flexneri*_MZS_191, GenBank accession no: MW774908.1) through electrophoresis in 1.5% agarose gels with ethidium bromide.

## Statistical analysis

Statistical analysis were done by SPSS.

## Result

### Isolation and identification of *Shigella spp*. through selective plating and TSI

The characteristic pink colonies (Fig 1A) on XLD (Xylose Lysine Deoxycholate) medium were suspected for *Shigella* and *S. flexneri*.Through direct stool culture on the XLD agar plate, 107 samples out of 204 showed pink colonies. In pure culture the pink colonies formed transparent colonies (Fig 1B) on MacConkey agar plate. Thesetransparent pure colonies were used for biochemical test TSI.

Red/acidic slant and yellow/basic butt indicated *Shigella* (n = 29) (Fig 1C). By selective platting, and biochemical test-TSI, 71 *Salmonella paratyphi*, 2 *Salmonella typhi* and 5 *Pseudomonas aeruginosa* were suspected.

### Identification of *Shigella spp*. through *ipaH* gene amplification

The *ipaH* gene was identified in all suspected *Shigella* isolates. *S. flexneri*strain MZS-191 (GenBank Accession No. MW774908.1) was used as positive control (Fig 2A).

### Identification of *Shigella flexneri*through *rfc* gene amplification

This*rfc* gene was detected in 27 samples among 29 suspected *Shigella flexneri*. *S. flexneri*strain MZS-191 (GenBank Accession No. MW774908.1) was used as positive control (Fig 2B).

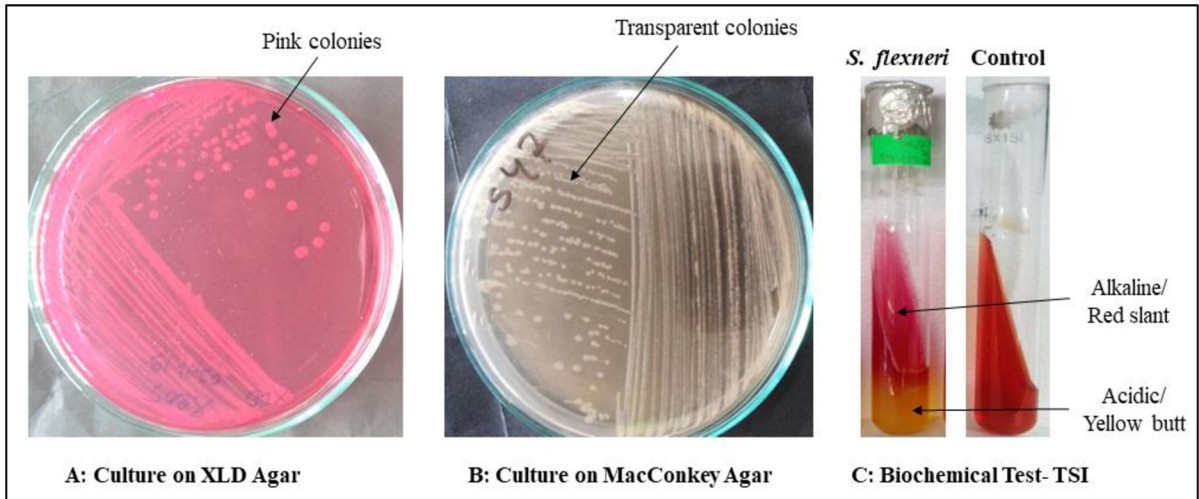

**Fig 1. Selective plating and biochemical test.** A: Target bacterial screening through XLD agar medium. Pink colonies were suspected as *Shigella*. B: Selected pink colonies were cultured on MacConkey agar medium and transparent colonies were selected for biochemical test TSI. C: On TSI biochemical test, acidic slants and alkaline butts without bubble were suspected as *Shigella spp*. and *Shigella flexneri*.

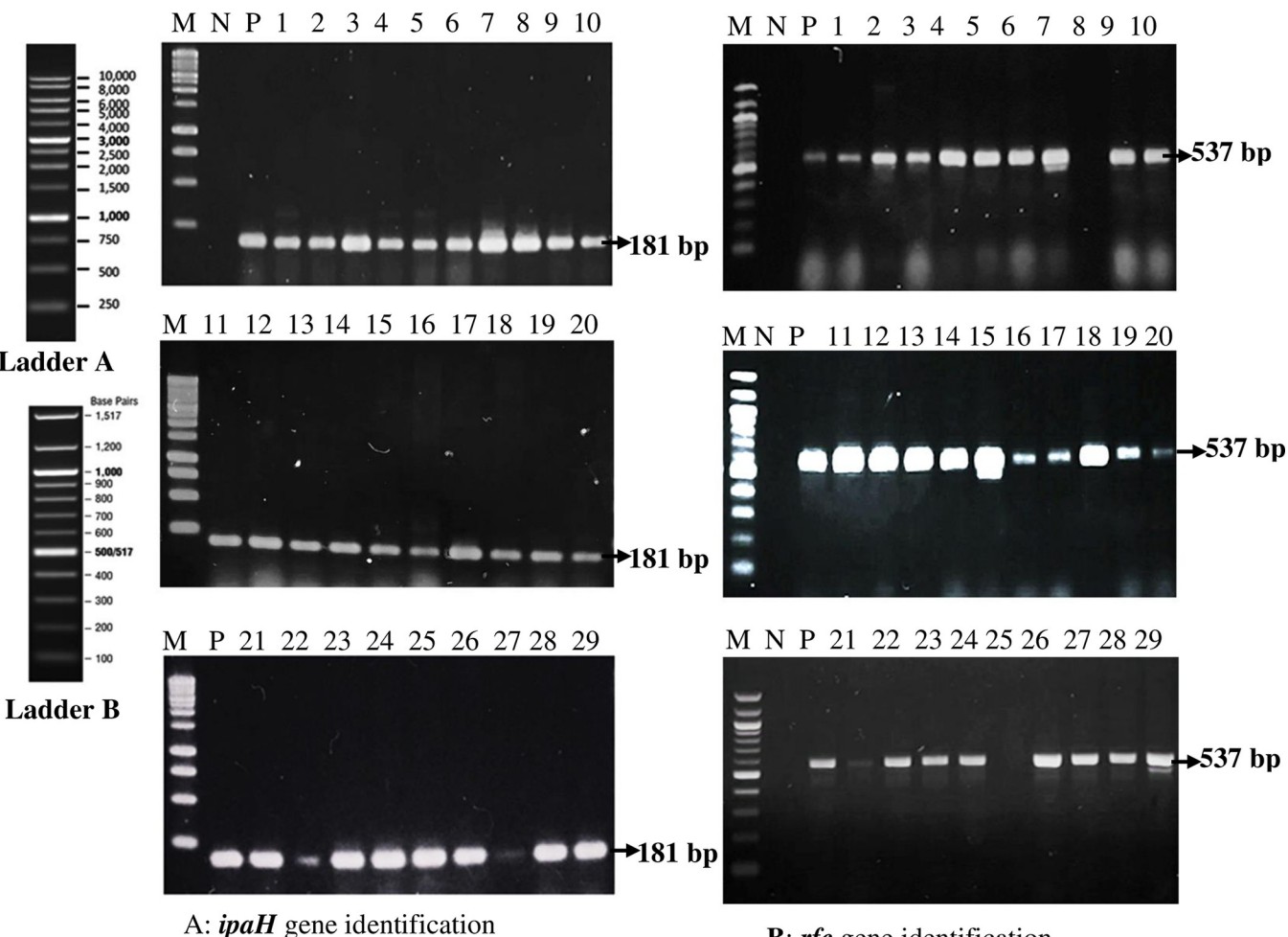

**Fig 2. Molecular identification of *Shigella spp.*and *Shigella flexneri*.** The study used PCR with specific primers to amplify the *ipaH* and *rfc* genes from suspected *Shigella flexneri* and *Shigella spp.* samples. The PCR products were analyzed through agarose gel electrophoresis, and a positive control *S. flexneri* MZS-191 (GenBank Accession No. MW774908.1) was included in both cases. Lane 1–29 represents clinical samples (*Shigella flexneri* except Lane 8 & 25 which represents *Shigella spp.*). A: The results revealed that all 29 suspected samples contained the *ipaH* gene (amplicon size ≈181 bp), confirming their identification as *Shigella flexneri* or *Shigella spp*. B: Furthermore, the *rfc* gene (amplicon size ≈ 537 bp), which is specific to *S. flexneri*, was detected in 27 out of 29 *ipaH*-containing samples, confirming their identification as *S. flexneri*. In contrast, samples MZS-9 and MZS-25 did not show the presence of the *rfc* gene, indicating that they belonged to other *Shigella* species. [P for Positive Control, N for Negative Control, M for DNA Marker/Ladder].

## Occurrence of diarrheal infections and Shigellosis

We obtained some most interesting data through selective plating, biochemical tests and molecular identifications.Different organisms were identified in diarrheal pediatric patients (Table 2).

We found that 14.2% (Fig 3A) diarrheal case was shigellosis and boys were more likely to be infected than girls (Fig 3B). In shigellosis cases, 93% (n = 27) were caused by *S. flexneri* and 7% (n = 2) were caused by other *Shigella spp.* (Fig 3C).

Our findings also showed that all of shigellosis patients except one were less than five years old. In this rang of age intervals, the maximum infant (n = 17) were 7months to one years old. Three infants of shigellosis were 1 day to 6 months (Fig 4B).

Not only Shigellosis, other diarrheal pediatric patients mostly were in 7 months to 1 year of age range (Fig 4B).

**Table 2. Occurrence of *Shigella flexneri* and *Shigella spp.* among diarrheic children under-5 years of age in Chattogram, Bangladesh.**

| Variables | | Frequency | Percent (%) | Sex | | Significance |
|---|---|---|---|---|---|---|
| | | | | Male | Female | |
| Microorganisms | *Shigella spp* | 2 | 1 | 0 | 2 | p<0.01 |
| | *Shigella flexneri* | 27 | 13.2 | 12 | 15 | |
| Shigellosis | Negative | 175 | 85.8 | 110 | 65 | p<0.01 |
| | Positive | 29 | 14.2 | 12 | 17 | |
| Ages | 1D ≤ 6M | 32 | 15.7 | 20 | 12 | p<0.01 |
| | 7M ≤ 1Y | 94 | 46.1 | 59 | 35 | |
| | 1Y 1M ≤ 5Y | 72 | 35.3 | 45 | 27 | |
| | 5Y < | 6 | 2.9 | 3 | 3 | |

*p values and level of significance were calculated by SPSS data analysis tool.

## Discussion

Shigellosis is a highly communicable bacterial diarrheal disease that is known to occur sporadically [24]. It is particularly concerning in low-income countries, where it can sometimes break out and cause epidemics and endemics, leading to a significant public health burden [1]. Shigellosis is caused by *Shigella spp.*, with *Shigella flexneri* being the most prevalent in developing countries [13,25].The highest rates of diarrheal disease are reported in developing countries like Bangladesh, where socioeconomic and behavioral factors contribute to the emergence and spread of multi-drug resistant strains of *S. flexneri*, which is a major concern [25]. These strains can be difficult to treat, leading to prolonged illness, and can contribute to the spread of the disease. Diagnosing shigellosis can be challenging using traditional culture methods, which can take anywhere from two to seven days after sampling and are not always successful [2]. As a result, proper and rapid diagnosis is critical to managing outbreaks and controlling the spread of the disease.

The prevalence of *Shigella spp.* as enteric pathogens with a global impact has been on the rise in recent years. This study was conducted to investigate the prevalence of *Shigella spp.* in diarrheal patients using cultural and biochemical methods. This study aimed to isolate *Shigella spp.* from diarrheal patients using cultural and biochemical methods, with confirmation of the

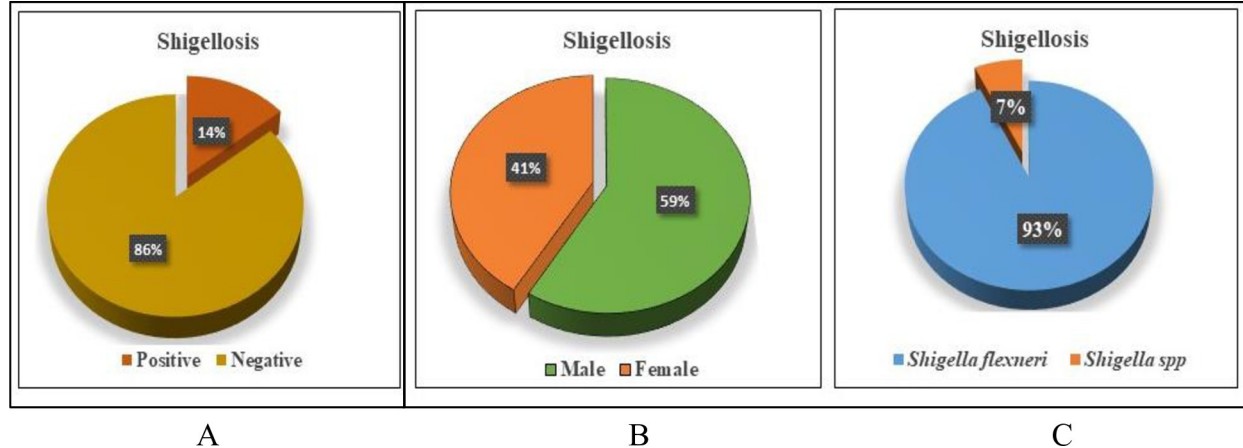

**Fig 3. Occurrence of Shigellosis and demographic characteristics of affected patients.** A: Among 204 diarrheoal patients, 14.2% (n = 29) cases were Shigellosis. B: Most of patients were male children and C: Among Shigellosis children, 93% (n = 27) were infected by *Shigella flexneri*.

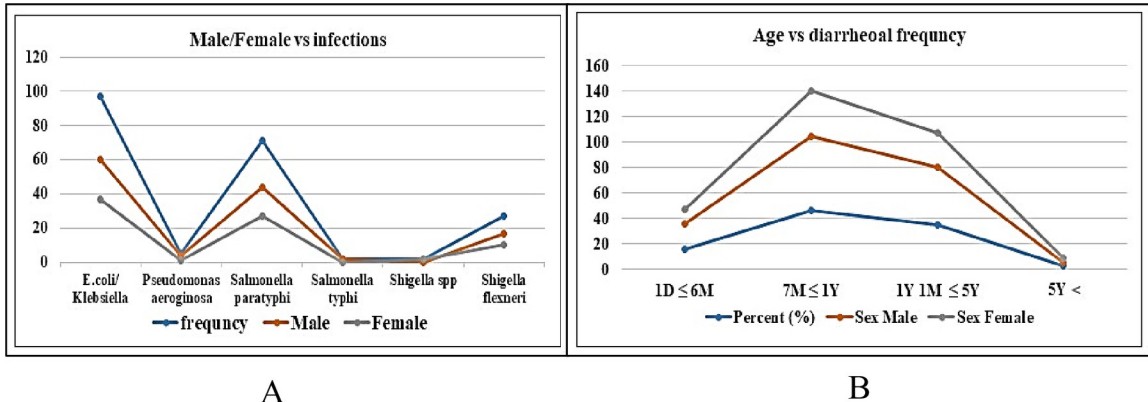

**Fig 4. The occurrence frequency of different organisms in diarrheal patients based on the age and gender of male and female children in different age groups.** A: The graph shows that out of 204 children with diarrhea, the majority of the identified organisms were *E. coli/Klebsiella* (n = 97), *Salmonella paratyphi* (n = 71), *Shigella flexneri* (n = 27), *Pseudomonas aeruginosa* (n = 5), *Shigella spp* (n = 2) and *Salmonella typhi* (n = 2). All of these bacteria are known to cause diarrheal diseases in the host and are considered pathogenic. It was observed that the prevalence of infection was higher among male children than female children. B: The majority of the infected children were between 7 months and 1yearold.

isolation rate through amplification of the *ipaH* gene. We found that 14.2% of diarrheal patients were infected with *Shigella*, which is consistent with rates found in studies conducted in Iran (14%) and Brazil (10%), but different from rates in studies conducted in India (4%) and Ethiopia (4%) [26]. In this study, the biochemical and the rapid, specific molecular identification of *Shigellaspp*.and *S.flexneri* were demonstrated. From 107 non-fermented bacterial cultures obtained through selective plating, 29 were identified as Shigella using the TSI biochemical test. However, the molecular PCR test confirmed that all the suspected bacteria were indeed *Shigella*. While TSI proved to be a reliable test for detecting *Shigella* in general, it was not entirely effective in accurately detecting *S. flexneri*, which was the specific focus of this study. These results emphasize the need for the use of molecular techniques in accurately and rapidly identifying both *Shigellaspp*.and *S. flexneri* in clinical and public health settings.In that case, 93% of the *S. flexneri* could be detected. However, specific *S. flexneri* detection requires molecular tests. As a result, we observed that molecular identification is the most dependable method for the identification of *S. flexneri*.For molecular identification of Shigellosis,we focused on the multi-copied pathogenic gene- *ipaH*, which is found on both chromosomes and plasmids of *Shigella* [20] and *EnteroinvasiveEscherichia coli* (*EIEC*) [21]. Non-temperature regulated IpaH protein, is also distinct from other invasive proteins such as IpaA, IpaB, IpaC, and IpaD both immunologically and at the DNA level. This *ipaH* gene is therefore an excellent target for molecular identification of *Shigella* [20,21]. We found this gene in all of the isolates. This finding indicates that *ipaH* is an important tool in the identification of shigellosis, allowing us to distinguish *Shigella* from non-fermented bacteria. This molecular method allows diagnosis centers to detect shigellosis with the greatest accuracy.The study conducted by Pholwat and colleagues in 2022 demonstrated that PCR assay amplifying the *ipaH* gene accurately identifies *Shigellaspp*. [20]. These results are consistent with our findings.Despite the higher cost associated with this method, it can be beneficial for the accurate identification of shigellosis, leading to appropriate treatment for patients. The investigation carried out by Jin Yang et al. 2020 [3] has confirmed that conventional PCR is an extremely convenient and effective technique for the detection of *Shigella spp*. from both genomic DNA samples and cultured bacterial strains. This method can detect the presence of *Shigella spp*. at very low levels, as low as $10^{-3}$ ng of genomic DNA or $10^2$ colony-forming units (cfu) of cultured bacteria [3]. For

specific identification of *S. flexneri*, we focused on the *rfc* gene that encodes the O-antigen polymerase responsible for polymerizing lipopolysaccharide chains. This gene is exclusive to the genome of *S. flexneri*, making it an effective tool for specific identification of this bacterial strain [12]. Through this gene identification, we found 27 *Shigella flexneri* out of 29 *Shigella* isolates. According to this data, *S. flexneri* was found to be the most responsible agent for Shigellosis in our country. This kind of findings were also reported by P. Parajuli *et al*. in 2019 [27]. According to our findings *S. flexneri* was responsible for 93% (n = 27) of shigellosis in Chattogram, Bangladesh.

After the proper identification of *S. flexneri*, we assessed the prevalence of Shigellosis. Our findings revealed that out of 29 children diagnosed with shigellosis, 28 were under the age of five, with a statistically significant difference (p<0.01) (Table 2). An even more concerning finding from our study was that out of the 28 children with shigellosis, 17 (58.62%) were between the ages of 7 months to 1 year (Table 2). This study also revealed an intriguing observation that the incidence of shigellosis and other diarrheal cases was higher among boys compared to girls. In addition, the identification and the prevalence of Shigellosis, we also investigated the other causes of diarrhea. Maximum diarrhea cases (47.5%, n = 97) were caused due to *E.coli/Klebsiella* infection (S1 Table). *Salmonella paratyphi* also causes a majority cases (34.8%) of diarrhea. In diarrheoal pediatric patients, *Salmonella typhi*, *Pseudomonas aeruginosa* were also identified. It is expected that this study will aid in precise identification of *Shigella spp*. and *S. flexneri* among patients with diarrhea at the molecular level, which will assist in the appropriate treatment of Shigellosis.

This study was limited by inadequate funding, leading to a small sample size and restricted geographical area, which limits the generalizability of the findings. Patient follow-up was also not possible, hindering a comprehensive understanding of the long-term effects of Shigellosis. Future studies should prioritize securing adequate funding to expand the research scope, explore the disease's prevalence in different regions, age groups, and risk factors, and investigate new treatment strategies, including drug resistance patterns. Such studies could inform effective prevention and treatment strategies for Shigellosis.

## Conclusion

The precise and reliable identification of shigellosis is crucial in the effective management of this infectious disease, particularly in cases caused by *Shigella flexneri* strain. The proper identification of shigellosis through reliable testing methods enables healthcare centers to promptly initiate appropriate treatment measures, thereby reducing the risk of severe complications and mortality. This study successfully identified shigellosis through molecular techniques and it is expected that this will assist in the appropriate treatment of Shigellosis. The study also revealed that children under the age of 5 were more vulnerable to shigellosis, with a higher incidence among boys than girls. In Chattogram, Bangladesh, *Shigella flexneri* was found to be responsible for 93% of shigellosis cases, highlighting the significance of identifying and managing this particular strain.

## Supporting information

**S1 Raw images.**
(PDF)

**S1 Table. Occurrence** *of E.coli/ Klebsiella*, *Pseudomonas aeruginosa*, *Salmonella paratyphi*, *Salmonella typhi*, *Shigella spp*, *and Shigella flexneri* **among diarrheic male and female**

**children under-5 years of age in Chattogram, Bangladesh.**
(PDF)

## Acknowledgments

We express our gratitude to the faculty and staff of the Department of Genetic Engineering and Biotechnology at the University of Chittagong, Bangladesh, for their support during our study.

## Author Contributions

**Conceptualization:** A. K. M. Zakir Hossain, A. M. Masudul Azad Chowdhury.

**Data curation:** A. K. M. Zakir Hossain, Md. Zahid Hasan, Sohana Akter Mina, Nahid Sultana.

**Formal analysis:** A. K. M. Zakir Hossain.

**Funding acquisition:** A. M. Masudul Azad Chowdhury.

**Investigation:** A. K. M. Zakir Hossain, A. M. Masudul Azad Chowdhury.

**Methodology:** A. K. M. Zakir Hossain.

**Project administration:** A. M. Masudul Azad Chowdhury.

**Resources:** A. M. Masudul Azad Chowdhury.

**Supervision:** A. M. Masudul Azad Chowdhury.

**Validation:** A. K. M. Zakir Hossain, A. M. Masudul Azad Chowdhury.

**Visualization:** A. K. M. Zakir Hossain.

**Writing – original draft:** A. K. M. Zakir Hossain, A. M. Masudul Azad Chowdhury.

**Writing – review & editing:** A. K. M. Zakir Hossain, Md. Zahid Hasan, Sohana Akter Mina, Nahid Sultana, A. M. Masudul Azad Chowdhury.

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
