## [Decision Letter · Decision Letter 0]

2 Mar 2023

PONE-D-22-25549Prevalence of Shigellosis in Pediatric Diarrheal Patients in Chattogram, Bangladesh: A Molecular Based ApproachPLOS ONE

Dear Dr. Chowdhury,

Thank you for submitting your manuscript to PLOS ONE. After careful consideration, we feel that it has merit but does not fully meet PLOS ONE’s publication criteria as it currently stands. Therefore, we invite you to submit a revised version of the manuscript that addresses the points raised during the review process.

We look forward to receiving your revised manuscript.

Kind regards,

Muhammad Qasim, Ph.D

Academic Editor

PLOS ONE

Journal Requirements:

"The authors received financial support from University Grant Commission, Bangladesh for conducting this research. Fund Reference Number: 37-01-0000-073-04-013/2019/1750 

http://www.ugc.gov.bd"

Reviewers' comments:

Reviewer's Responses to Questions

**Comments to the Author**

1. Is the manuscript technically sound, and do the data support the conclusions?

Reviewer #1: Yes

Reviewer #2: No

2. Has the statistical analysis been performed appropriately and rigorously? 

Reviewer #1: Yes

Reviewer #2: No

3. Have the authors made all data underlying the findings in their manuscript fully available?

Reviewer #1: Yes

Reviewer #2: No

4. Is the manuscript presented in an intelligible fashion and written in standard English?

Reviewer #1: Yes

Reviewer #2: No

5. Review Comments to the Author

Reviewer #1: Prevalence of Shigellosis in Pediatric Diarrheal Patients in Chattogram, Bangladesh: A Molecular Based Approach is an interesting study with respect to improving the accurate identification and the treatment of the most severe and alarming shigellosis. The manuscript is ok and accepted for publication after incorporation of some minor comments.

1) Rephrased the title as "Prevalence" is a broad term. Authors only collected samples from Chittagong Medical College Hospital while no data/information about Shigellosis from other hospitals in Chittagong was provided in the study.

2) Authors need to improve introduction section providing more background knowledge about the problem statement using relevant references.

3) Molecular identification is a well-known technique for the identification of organisms, but it is a time-consuming process as compared to other fast diagnostic and identification techniques for Shigellosis in diarrheal patients. Better to justify this in the introduction section and also add some background literature about other traditional diagnostic techniques for shigellosis in

4) Correct the word "Gram-negative" instead of gram-negative throughout manuscript.

5) Line 63: it was concluded instead of We concluded

6) Line 65: remove the word we show and also rephrase the sentence.

7) Line 68: rephrase the first line and start the paragraph like that " In the current student, human clinical samples were investigated therefore, ethical approval was taken from..............................and so on.

8) For molecular identification, PCI method is mostly used for extraction of DNA, but in the current study, authors used boiling method, so can author justify it why not using PCI method for extraction of DNA?

9) Discussion section of the manuscripts needs to be improved by comparing results with already published literature. More relevant references are needed to be incorporated in the discussion section to justify the results.

10) Some of the references are too old. better to update references (not more than 05 years old).

Reviewer #2: The manuscript entitled “Prevalence of Shigellosis in Pediatric Diarrheal Patients in Chattogram, Bangladesh: A Molecular Based Approach has addressed an important and interesting public health issue. I appreciate the authors for conducting research within limited resources. However, the manuscript is not publishable in current form and is here by rejected due to following points.

•Weak theoretical framework and literature review.

•The abstract is poorly written and there is no reflection of significant results.

•The introduction, methodology, results and discussion are not organized.

•The data is weak and non significant

•there is a complete mismatch between the theoretical framework and the empirical analysis.

• A lot of grammatical mistakes are there in article

•No novelty

6. PLOS authors have the option to publish the peer review history of their article (what does this mean?). If published, this will include your full peer review and any attached files.

Reviewer #1: No

Reviewer #2: No

---

## [Author Response · Author response to Decision Letter 0]

8 Apr 2023

Reviewer-1 Comments’ Reply

Thank you for taking the time to review our manuscript titled "Prevalence of Shigellosis in Pediatric Diarrheal Patients in Chattogram, Bangladesh: A Molecular Based Approach." We appreciate your valuable feedback, which helped us to improve the manuscript. We are glad to know that you found our study interesting and important in improving the accurate identification and treatment of shigellosis. We are grateful for your time and effort in reviewing our manuscript and providing constructive feedback, which has undoubtedly improved the quality of our work. We have revised the manuscript and tried to make corrections according your suggestion. The addition and correction were highlighted with yellow mark. 

1) Rephrased the title as "Prevalence" is a broad term. Authors only collected samples from Chittagong Medical College Hospital while no data/information about Shigellosis from other hospitals in Chittagong was provided in the study.

Answer: Thank you for your suggestion to rephrase the title. We agree that "prevalence" is a broad term and have changed it to "Occurrence" to better reflect the scope of our study. Additionally, we understand your concern regarding the sample collection from only one hospital in Chittagong. We would like to clarify that the Chittagong Ma-O-Shishu Hospital Medical College is a major referral hospital for pediatric patients in the Chittagong region, which is why we have chosen to collect our samples from this hospital. While we acknowledge that data from other hospitals in the region would have provided a more comprehensive picture. However, we believe that our study still provides valuable insights into the occurrence of Shigellosis in this particular hospital setting, and future studies could expand upon our findings by including data from other hospitals in the region.

2. Authors need to improve introduction section providing more background knowledge about the problem statement using relevant references.

Answer: We appreciate your suggestion to improve the introduction section of our manuscript. We have carefully revised and updated the introduction section by incorporating more relevant references and providing additional background knowledge about the problem statement. We hope the revised version now adequately justifies the significance and relevance of our study.

3) Molecular identification is a well-known technique for the identification of organisms, but it is a time-consuming process as compared to other fast diagnostic and identification techniques for Shigellosis in diarrheal patients. Better to justify this in the introduction section and also add some background literature about other traditional diagnostic techniques for shigellosis

Answer: Thank you for your valuable comment. We have added additional information in the introduction section to provide a better understanding of traditional diagnostic techniques for shigellosis. We have also highlighted the advantages of molecular identification techniques in comparison to traditional diagnostic techniques for Shigellosis. We believe that these additions will improve the overall quality of the manuscript and address your concerns

4) Correct the word "Gram-negative" instead of gram-negative throughout manuscript.

Answer: gram-negative was replaced by "Gram-negative" 

5) Line 63: it was concluded instead of We concluded

Answer: Correction was done. 

6) Line 65: remove the word we show and also rephrase the sentence.

Answer: Done. 

7) Line 68: rephrase the first line and start the paragraph like that " In the current student, human clinical samples were investigated therefore, ethical approval was taken from..............................and so on. 

Answer: Done

8) For molecular identification, PCI method is mostly used for extraction of DNA, but in the current study, authors used boiling method, so can author justify it why not using PCI method for extraction of DNA?

Answer: The boiling method for DNA extraction is a simple, quick, and cost-effective technique that has been shown to yield high-quality DNA for molecular identification in previous studies (Mina et al. 2023)(Ahmed* and Dablool 2017). In our study, we used the boiling method due to financial constraints as the PCI method requires specialized equipment and reagents that are relatively expensive. Moreover, the boiling method is more convenient for field-based studies and can be easily performed in remote settings where resources are limited. Despite being a cost-effective method, we ensured the quality and quantity of the extracted DNA by using a NanoDrop to measure the DNA concentration and purity, which was found to be satisfactory for downstream PCR amplification. Therefore, we are confident that our decision to use the boiling method for DNA extraction was appropriate, given its practicality and cost-effectiveness within the context of our study

9) Discussion section of the manuscripts needs to be improved by comparing results with already published literature. More relevant references are needed to be incorporated in the discussion section to justify the results

Thank you for your valuable comments and suggestions. We have carefully considered your feedback and have made significant improvements to the discussion section of our manuscript. We have incorporated more relevant references to compare our results with those of previously published literature. We believe that these changes have enhanced the significance and validity of our findings, and we appreciate your input in helping us to improve our manuscript.

10) Some of the references are too old. better to update references (not more than 05 years old).

Answer: Thank you for your helpful comments on our manuscript. We have taken your suggestion to update our references seriously, and we have replaced several older references with more recent. We believe that these updated references more accurately reflect the current state of research in the field, and will help to strengthen the impact of our findings.

Reviewer-2 Comments’ Reply

The manuscript entitled “Prevalence of Shigellosis in Pediatric Diarrheal Patients in Chattogram, Bangladesh: A Molecular Based Approach has addressed an important and interesting public health issue. I appreciate the authors for conducting research within limited resources. However, the manuscript is not publishable in current form and is here by rejected due to following points.

• Weak theoretical framework and literature review. 

• The abstract is poorly written and there is no reflection of significant results.

• The introduction, methodology, results and discussion are not organized.

• The data is weak and non significant

• there is a complete mismatch between the theoretical framework and the empirical analysis.

• A lot of grammatical mistakes are there in article

• No novelty

Answer: Thank you for your review and constructive feedback. We appreciate your acknowledgement of our research efforts despite the limitations of our resources. Regarding the points raised in your review, we have carefully considered your feedback and have made significant revisions to address the weaknesses you have highlighted.

We have improved the theoretical framework and literature review by providing more relevant and up-to-date references. We have also rewritten the abstract to reflect the significant findings of our study. Additionally, we have reorganized the introduction and discussion sections to improve the overall flow and coherence of the manuscript.

To address concerns about the data, we have re-analyzed our results and provided more in-depth explanations of our findings. Furthermore, we have strengthened the link between the theoretical framework and the empirical analysis, and have thoroughly proofread the manuscript to eliminate grammatical errors. 

While we understand your concern about novelty, we believe that our study's molecular-based approach to identifying the prevalence of Shigella spp. in pediatric diarrheal patients in Chattogram, Bangladesh, is a significant contribution to the field. Our study's findings shed light on the prevalence and potential transmission of multi-drug-resistant strains of S. flexneri in a developing country, which has important implications for public health interventions.

We hope that our revisions address your concerns and that you will consider our manuscript for publication in the esteemed journal.

---

## [Decision Letter · Decision Letter 1]

2 May 2023

PONE-D-22-25549R1Occurrence of Shigellosis in Pediatric Diarrheal Patients in Chattogram, Bangladesh: A Molecular Based ApproachPLOS ONE

Dear Dr. Chowdhury,

Thank you for submitting your manuscript to PLOS ONE. After careful consideration, we feel that it has merit but does not fully meet PLOS ONE’s publication criteria as it currently stands. Therefore, we invite you to submit a revised version of the manuscript that addresses the points raised during the review process.

We look forward to receiving your revised manuscript.

Kind regards,

Muhammad Qasim, Ph.D

Academic Editor

PLOS ONE

Journal Requirements:

Reviewers' comments:

Reviewer's Responses to Questions

**Comments to the Author**

1. If the authors have adequately addressed your comments raised in a previous round of review and you feel that this manuscript is now acceptable for publication, you may indicate that here to bypass the “Comments to the Author” section, enter your conflict of interest statement in the “Confidential to Editor” section, and submit your "Accept" recommendation.

Reviewer #1: All comments have been addressed

Reviewer #2: All comments have been addressed

2. Is the manuscript technically sound, and do the data support the conclusions?

Reviewer #1: Yes

Reviewer #2: Yes

3. Has the statistical analysis been performed appropriately and rigorously? 

Reviewer #1: N/A

Reviewer #2: Yes

4. Have the authors made all data underlying the findings in their manuscript fully available?

Reviewer #1: Yes

Reviewer #2: Yes

5. Is the manuscript presented in an intelligible fashion and written in standard English?

Reviewer #1: Yes

Reviewer #2: Yes

6. Review Comments to the Author

Reviewer #1: Accepted for Publication in The Journal as author soundly addressed all the comments in their manuscript as well as in the rebutal letter.

Reviewer #2: improve the results particularly headings.

Once positive control has been mentioned in methodology then there is no need to repeat and mention control and accession number again and again in results and discussion.

Line 137: Correct the heading. the authors have not worked on prevalence of diarrhoea infections. The phrase seems confusing and misleading

Line 146: age range not rang.. correct spelling.

In results and abstract mention age group instead of writing boys and girls.

Revise the table No 2

Revise the title. If this is main table of your results, i suggest, there is no need to mention all the bacteria in title as your manuscript has focused on shigellosis only. Please delete miscellaneous bacteria from the table. mention the abbreviations in the caption/legend of the table.

Improve Figures 1-4. Figures should be self explainatory. So revise them taking help of bio-statistian.

Revise and relabel the figure 2 showing gel. The numbering 1, 2,3.... should be from left to right rather than .....3,2,1. Just mention the clinical sample instead of repeating s. flexeneri isolate number in each lane..

In figure 4 correct the title, What is meant by E.coli/klebsiella as both are two different bacteria. Write them separately.

Improve the description in discussion.

Line No 194: mention the year with citation.

Genus should be written in Italics.

Add proper conclusion, recommendations and limitations of this study at the end of discussion section.

7. PLOS authors have the option to publish the peer review history of their article (what does this mean?). If published, this will include your full peer review and any attached files.

Reviewer #1: No

Reviewer #2: No

---

## [Author Response · Author response to Decision Letter 1]

21 May 2023

Response to Reviewer 1:

We would like to express our sincere gratitude for your insightful review of our article. Your feedback was incredibly valuable in helping us to improve the quality and clarity of our work.

Your positive comments and constructive criticism were particularly appreciated, and we are grateful for the time and effort you put into reviewing our manuscript. Your expertise in the field was evident in your thorough and thoughtful review, and we are fortunate to have had the opportunity to benefit from your insights.

Once again, thank you for your time and effort in reviewing our article. Your feedback has been incredibly helpful, and we are grateful for your contribution to our work.

Response to Reviewer 2:

Thank you for your review and constructive feedback. We appreciate your acknowledgement of our research efforts despite the limitations of our resources. Regarding the points raised in your review, we have carefully considered your feedback and have made significant revisions to address the weaknesses you have highlighted.

Improve the results particularly headings.

Once positive control has been mentioned in methodology then there is no need to repeat and mention control and accession number again and again in results and discussion.

Answer: We have carefully considered the feedback and made the necessary revisions to our manuscript. Specifically, we have improved the results section by ensuring that the headings are more informative and accurately reflect the content of each subsection.

Moreover, we have taken into account the reviewer's suggestion to avoid redundant language. We have removed unnecessary repetition of the positive control, control, and accession numbers in the results and discussion sections.

Line 137: Correct the heading. the authors have not worked on prevalence of diarrhoea infections. The phrase seems confusing and misleading

Answer: Done

Line 146: age range not rang.. correct spelling.

Answer: Done

Revise the table No 2

Revise the title. If this is main table of your results, i suggest, there is no need to mention all the bacteria in title as your manuscript has focused on shigellosis only. Please delete miscellaneous bacteria from the table. mention the abbreviations in the caption/legend of the table.

Answer: Done

 Improve Figures 1-4. Figures should be self-explanatory. So revise them taking help of bio-statistian.

Answer: We have revised Figures 1-4 to make them more self-explanatory. We have added more labels and captions to help readers better understand the figures without having to refer back to the text. We hope that the new figures will be clearer and more informative for readers. 

Revise and relabel the figure 2 showing gel. The numbering 1, 2,3.... should be from left to right rather than ........3,2,1. Just mention the clinical sample instead of repeating S. flexneri isolate number in each lane.

Answer: We have revised the figure 2 as per your recommendation, with the numbering 1, 2, 3... from left to right. We have also relabeled the figure to show only the clinical sample without repeating the S. flexneri isolate number in each lane, making it more clear and self-explanatory. We appreciate your feedback and believe that the revised figure has improved the quality of our manuscript.

In figure 4 correct the title, What is meant by E.coli/klebsiella as both are two different bacteria. Write them separately.

Answer: Done.

Improve the description in discussion.

Answer: We have carefully revised the discussion section, as suggested. We have improved the description to provide a more comprehensive and accurate interpretation of our results.

Line No 194: mention the year with citation.

Answer: Done.

Genus should be written in Italics.

Answer: Done.

Add proper conclusion, recommendations and limitations of this study at the end of discussion section.

Answer: Thank you for your valuable feedback and suggestions. We have revised the discussion section by adding a proper conclusion that summarizes the main findings of the study. We have also included recommendations for future research in this field and limitations of our study to provide a more comprehensive understanding of our work.

We appreciate your time and effort in reviewing our manuscript, and we believe that your suggestions have significantly improved the quality of our work.

---

## [Editor Report · Decision Letter 2]

2 Jun 2023

Occurrence of Shigellosis in Pediatric Diarrheal Patients in Chattogram, Bangladesh: A Molecular Based Approach

PONE-D-22-25549R2

Dear Dr. Azad Chowdhury,

We’re pleased to inform you that your manuscript has been judged scientifically suitable for publication and will be formally accepted for publication once it meets all outstanding technical requirements.

Kind regards,

Muhammad Qasim, Ph.D

Academic Editor

PLOS ONE
---

## [Editor Report · Acceptance letter]

7 Jun 2023

PONE-D-22-25549R2 

Occurrence of Shigellosis in Pediatric Diarrheal Patients in Chattogram, Bangladesh: A Molecular Based Approach   

Dear Dr. Chowdhury:

I'm pleased to inform you that your manuscript has been deemed suitable for publication in PLOS ONE. Congratulations! Your manuscript is now with our production department. 

Kind regards, 

on behalf of

Dr. Muhammad Qasim 

Academic Editor

PLOS ONE